# Determinants of Acceptance of COVID-19 Vaccination in Healthcare and Public Health Professionals: A Review

**DOI:** 10.3390/vaccines11020311

**Published:** 2023-01-31

**Authors:** Fathema Ghare, Rehab Meckawy, Michael Moore, Marta Lomazzi

**Affiliations:** 1World Federation of Public Health Associations (WFPHA), Institute of Global Health, University of Geneva, Campus Biotech—G6, Chemin des Mines 9, 1202 Geneva, Switzerland; 2Public Health and Community Medicine Department, Alexandria Faculty of Medicine, Alexandria University, Champollion Street, Al Attarin, Alexandria 21321, Egypt; 3Institute of Global Health, University of Geneva, Campus Biotech—G6, Chemin des Mines 9, 1202 Geneva, Switzerland

**Keywords:** attitudes, COVID-19, health care personnel, sentiment, vaccine, vaccine hesitancy, public health professionals

## Abstract

Vaccinations of healthcare workers (HCWs) aim to directly protect them from occupational diseases, and indirectly protect their patients and communities. However, studies increasingly highlight that HCWs can be vaccine hesitant. This review aims to analyze HCWs’ and public health professionals’ sentiments toward COVID-19 (Coronavirus Disease 2019) vaccination and determinants across different countries. A search strategy was conducted in PubMed using keywords such as “COVID-19”, “sentiment/acceptance”, “healthcare workers”, “vaccine hesitancy”, and “influenza”. A total of 56 articles were selected for in-depth analyses. The highest COVID-19 vaccination uptake was found in an Italian study (98.9%), and the lowest in Cyprus (30%). Older age, male gender, the medical profession, higher education level, presence of comorbidities, and previous influenza vaccination were associated with vaccine acceptance. Factors for low acceptance were perceived side effects of the vaccine, perceived lack of effectiveness and efficacy, and lack of information and knowledge. Factors for acceptance were knowledge, confidence in the vaccine, government, and health authorities, and increased perception of fear and susceptibility. All studies focused on healthcare providers; no studies focusing on public health professionals’ sentiments could be found, indicating a gap in research that needs to be addressed. Interventions must be implemented with vaccination campaigns to improve COVID-19 vaccine acceptance.

## 1. Introduction

On 30 January 2020, the World Health Organization (WHO) declared the outbreak of the SARS-CoV-2 virus a public health emergency of international concern and a pandemic on 11 March 2020. As of 30 November 2022, the infection has resulted in a total of 643.18 million cases and 6.64 million deaths globally [1]. The COVID-19 (Coronavirus Disease 2019) pandemic has had severe social, health, and economic implications, and has put considerable strain on health systems. To prevent further morbidity and mortality, the COVID-19 vaccines were developed rapidly and rolled out in many countries in 2020–2021 to control the pandemic. By 28 December 2022, 69.1% of the world population has received at least one dose of the vaccine and 25.9% of people in low-income countries have received at least one dose [2].

Vaccines are the most effective intervention in controlling vaccine-preventable diseases. However, they have been increasingly scrutinized due to misinformation and social media platforms, resulting in low vaccine acceptance [3,4]. According to the SAGE (Strategic Advisory Group of Experts on Immunization) working group of the WHO, vaccine hesitancy is defined as “the delay in acceptance or refusal of vaccination despite the availability of vaccination services” [5]. It exists on a continuous spectrum between complete refusal and complete acceptance [5]. In 2019, the WHO identified low vaccine acceptance and the risk of a pandemic as two of the ten threats to global health [6]. As the risk of emerging variants remains, low acceptance of the COVID-19 vaccines is a threat to public health and is a factor in prolonging the pandemic until global herd immunity improves.

With healthcare workers (HCWs) at the frontlines of the pandemic, they are at a high risk of acquiring nosocomial infections, and vaccinating them against the COVID-19 infection is of the highest priority as recommended by the WHO [7]. Protecting HCWs from the infection will reduce the risk to themselves, spread to close contacts, and prevent overwhelming health systems through increased staff absences. However, the rapid development of the vaccines has contributed to their low acceptance, fueled by concerns surrounding their safety and efficacy [8]. As patient advisors, negative sentiments about the COVID-19 vaccine among HCWs and public health professionals could affect sentiments within the general population [9]. Although public health professionals may not necessarily be in patient-facing roles, they are recognized as proponents of population health and immunization. Understanding factors that influence HCWs’ and public health professionals’ sentiments towards the COVID-19 vaccines facilitates broader understanding of their concerns, helps strategize ways to address them, and builds vaccine confidence.

In this literature review, we aim to investigate the determinants, factors, and barriers that influence COVID-19 vaccination sentiment in HCWs. We conducted a comparative study of the variations in the determinants and the factors across different countries and regions. In addition, we assess the impact of influenza vaccination history on COVID-19 vaccination. This review is part of a larger project conducted by the World Federation of Public Health Association (WFPHA) investigating how the pandemic has impacted the vaccination sentiment of HCWs.

## 2. Materials and Methods

A literature search was conducted on the PubMed database between 2–5 May 2022. A Boolean search strategy using a combination of pertinent MeSH terms and keywords was performed (Table A1), resulting in 299 articles from PubMed and one grey literature report. The search was restricted to include only articles that were published in the English language since the beginning of the pandemic. The terms “pregnant women*”, “pregnan*”, and “parent*” was used along with the “NOT” Boolean operator in the search strategy to exclude studies on pregnant women and parents, resulting in 255 hits. Abstracts and titles were then screened for relevance. Articles on the general population that solely denote HCWs as an occupation, on students studying in healthcare, and on patients were excluded. Studies focusing on the impact of the COVID-19 pandemic on influenza or other vaccine sentiments were also excluded. The articles that were included were peer-reviewed and focused on HCWs’ sentiments on COVID-19 vaccination and the factors, barriers, or determinants to COVID-19 vaccination. Grey literature through google advanced search was conducted through which a report by PAHO (Pan-American Health Organization) was included. The detailed steps of the screening and selection process are outlined in Figure 1. The major factors influencing the sentiment towards COVID-19 vaccination highlighted through the review are presented in the results.

Inclusion Criteria:Primary studies (cross-sectional, cohort, and longitudinal) and reviews (scoping reviews, systematic reviews, and meta-analysis) with access to the full text;Population: Healthcare workers or public health professionals;Outcome: Articles examining the sentiments of HCWs towards COVID-19 vaccines and their uptake;English language articles from any country.Exclusion Criteria:HCWs are solely listed as an occupational determinant of vaccine acceptance and there is no in-depth analysis of COVID-19 vaccination sentiments within this population;Focus is not on the sentiments of HCWs regarding the COVID-19 vaccines;Focus is on the influence of influenza on COVID-19 and not vice versa;Inaccessible articles, letters, and non-original research articles were excluded.

## 3. Results

### 3.1. Characteristics of Studies

A total of 56 articles were included in the final sample consisting of cross-sectional studies (n = 48), cohort studies (n = 2), systematic reviews (n = 2), scoping reviews (n = 2), systematic review and meta-analysis (n = 1), and a grey literature report [8,10,11,12,13,14,15,16,17,18,19,20,21,22,23,24,25,26,27,28,29,30,31,32,33,34,35,36,37,38,39,40,41,42,43,44,45,46,47,48,49,50,51,52,53,54,55,56,57,58,59,60,61,62,63,64]. The cross-sectional and cohort studies were from Greece (n = 6), China (n = 5), France (n = 4), Turkey (n = 4), Italy (n = 4), Cyprus (n = 3), UK (n = 3), Germany (n = 2), Lebanon (n = 2), Spain (n = 2), Egypt (n = 2), the US (n = 2), Azerbaijan, Albania, Czech Republic, India, Iraq, Israel, Kosovo, Palestine, Saudi Arabia, Singapore, Slovakia, Switzerland, Tunisia, and the UAE (Table 1) [10,11,12,13,14,15,16,17,18,19,20,21,22,23,24,26,27,28,30,31,32,33,36,37,38,39,40,41,42,43,44,45,46,47,48,49,50,51,52,54,55,56,57,58,59,60,61,62,63,64]. The studied population in most articles were HCWs (n = 46), nurses (n = 6), dentists, midwives, pharmacists, pediatricians, and physicians [8,10,11,12,13,14,15,16,17,18,19,20,21,22,23,24,25,26,27,28,29,30,31,32,33,34,35,36,37,38,39,40,41,42,43,44,45,46,47,48,49,50,51,52,53,54,55,56,57,58,59,60,61,62,63,64]. Most studies were conducted in the European region (n = 27), the Middle East (n = 12), and Asia (n = 7) [11,12,13,14,15,17,19,20,21,22,24,26,27,30,31,33,36,37,38,39,40,41,42,43,44,45,46,47,48,49,50,51,52,54,55,56,57,60,61,62,63]. The highest COVID-19 vaccination uptake was found in an Italian cross-sectional study (98.9%) [15]. Although the lowest rate was found in a global scoping review (20.7%) among nurses, a cross-sectional study among nurses and midwives in Cyprus had the lowest vaccine acceptance rate (30%) [19,29].

### 3.2. Determinants of COVID-19 Vaccine Acceptance

Several demographic determinants were identified to be associated with vaccine acceptance as highlighted. Demographic determinants were identified, and the number of studies and their origins were characterized accordingly in Table 2.

#### 3.2.1. Gender

Studies have found that male HCWs are more inclined to receive the COVID-19 vaccines compared to female HCWs (n = 25/56) [8,10,16,18,21,23,24,25,26,29,34,35,36,37,38,42,43,44,46,49,52,54,59,61,63]. The highest number of articles were from within the European region (n = 9/27) from France (n = 3/5), Greece (n = 2/6), and Spain (n = 2/2) [21,36,38,43,46,49,54,63]. Middle Eastern studies (n = 6/12) were mostly from Turkey (n = 3/4), while otherstudies were from the UK (n = 2/3), and the US (n = 2/2) [10,16,23,24,26,37,44,52,59,61]. In addition, most reviews (n = 5/6) found similar results [8,25,29,34,35]. On the contrary, Cyprus, Greece, and Saudi Arabia found that female HCWs were more accepting [13,19,22].

#### 3.2.2. Age

Older HCWs (30–50 years) are more likely to accept vaccination (n = 24/56) as highlighted in studies mainly from Europe (n = 8/27) which includes France (n = 3/5), Cyprus (n = 2/3), Germany, Greece, Spain, and the Czech Republic [19,21,38,47,48,50,51,54]. This is followed by studies from other countries, such as Turkey (n = 3/4), and the UK (n = 2/3) [16,26,27,58,59,62]. In addition, five reviews and a multi-country study identified older age as a determinant [8,25,29,32,34,35]. HCWs of over 60 years of age were the most likely to accept a COVID-19 vaccine compared to other age groups in studies from the UK, Germany, Switzerland, and Azerbaijan [10,17,43,63]. Contrastingly, five articles from Italy, China, Egypt, and Palestine found that younger HCWs (18–<30) are more likely to accept COVID-19 vaccines [14,15,18,31,37]. Interestingly, the aforementioned Egyptian study reports higher acceptance among younger HCWs; however, it was not a predictor for COVID-19 vaccination [18].

#### 3.2.3. Occupation

Vaccine acceptance rates were the highest amongst physicians (n = 31/56) and were mainly from multiple studies within the European region (n = 15/27) in the following countries: Italy (n = 4/4), Greece (n = 3/6), and France (n = 3/5) [8,10,12,14,15,16,18,20,21,23,25,26,27,32,34,36,37,38,40,46,47,48,50,51,54,56,57,60,62,63,64]. Similar results were found for the US (n = 2/2), China (n = 2/5), Turkey (n = 2/4), three of six reviews, a report by PAHO, and a multi-country study [8,16,23,25,26,27,32,34,57,60,64]. In contrast, vaccine acceptance was lowest amongst nursing professionals, such as registered nurses, nursing associates, or auxiliary nurses (n = 18/56), within Europe (n = 7) in France (n = 2/5) and Italy (n = 2/2) [10,12,14,16,18,21,27,34,36,37,46,48,50,58,60,62,63,64]. Other multiple studies were from the UK (n = 2/2) and a systematic review [10,34,59].

#### 3.2.4. Ethnicity

Two reviews and three studies from the UK (n = 2) and the US identified ethnicity to be a determinant for COVID-19 vaccine acceptance [10,23,25,34,59]. Overall, studies report that Black HCWs were less accepting of the vaccines [10,25,34,59]. For instance, a study performed in the UK found that out of the 23% of HCWs that were vaccine hesitant, Black Caribbean (43.1%), Mixed White and Black Caribbean (59.5%), Black African (60.2%), Chinese (65.6%), Pakistani (68.3%), and White other ethnic groups (69.0%) reported lower acceptance than those from the White British group (77.7%) [59]. Furthermore, a UK-based study reports vaccination intent was significantly higher among Black HCWs compared to White, Indian, and other Asian HCWs [10]. Some studies have contrasting results for Asian HCWs. A study conducted in the US found that Asian HCWs were less likely to be accepting than non-Hispanic White HCWs [23]. In comparison, a rapid systematic review notes that Asians were more willing to be vaccinated compared to other groups [34].
vaccines-11-00311-t002_Table 2Table 2Overview of the number of studies and the countries that identified determinants associated with vaccine acceptance.DeterminantsTotal Studies (n)Country *Reviews/Reports/Multi-National StudiesAge


18–305China, Egypt, Italy, Palestine [14,15,18,31,37]
30–5024China, Cyprus, Czech Republic, France, Germany, Greece, India, Saudi Arabia, Spain, Turkey, UK, US [12,13,16,19,21,26,27,38,47,48,50,51,54,58,59,60,62]Biswas et al., Hajure et al., Khubchandani et al., Leigh et al., Li et al., Luo et al., PAHO [8,25,29,32,34,35,64]>604Azerbaijan, Germany, Switzerland, UK [10,17,43,63]
Gender


Male25Albania, Cyprus, Czech Republic, Egypt, France, Germany, Greece, Kosovo, Lebanon, Palestine, Spain, Switzerland, Turkey, UAE, UK, US [10,16,18,21,23,24,26,36,37,38,42,43,44,46,49,52,54,59,61,63]Biswas et al., Hajure et al., Khubchandani et al., Li et al., Luo et al. [8,25,29,34,35]Female3Cyprus, Greece, Saudi Arabia [13,19,22]
Education12China, Germany, Greece, Italy, Turkey, US [20,23,27,43,46,55,60]Biswas et al., Hajure et al., Khubchandani et al., Li et al., Shakeel et al. [8,25,29,34,53]Occupation


Nurses18China, Egypt, France, Germany, Greece, India, Israel, Italy, Palestine, Switzerland, Turkey, UK, US [10,12,14,16,18,21,27,36,37,46,48,50,59,60,62,63]Li et al., PAHO [34,64]Physicians31China, Cyprus, Czech Republic, Egypt, France, Germany, Greece, India, Israel, Italy, Palestine, Slovakia, Spain, Switzerland, Turkey, UK, US [10,12,14,15,16,18,20,21,23,26,27,36,37,38,40,46,47,48,50,51,54,56,57,60,62,63]Biswas et al., Hajure et al., Leigh et al., Li et al., PAHO [8,25,32,34,64]Ethnicity5UK, US [10,23,59]Hajure et al., Li et al. [25,34]Comorbidities9Azerbaijan, China, Egypt, Greece, US [17,18,23,36,57]Biswas et al., Hajure et al., Khubchandani et al., Li et al., [8,25,29,34]Previous Influenza Vaccination39Albania, Azerbaijan, China, Cyprus, Czech Republic, Egypt, France, Greece, Italy, Kosovo, Lebanon, Palestine, Slovakia, Spain, Switzerland, Tunisia, Turkey, UAE, UK, US [10,14,15,16,17,18,19,21,22,23,24,26,27,28,36,37,38,40,41,44,45,47,48,49,52,54,55,56,57,59,60,61,63]Biswas et al., Hajure et al., Khubchandani et al., Li et al., Luo et al., Shakeel et al. [8,25,29,34,35,53]Previous COVID-19 Infection16Azerbaijan, Albania, Cyprus, Czech Republic, Egypt, France, Greece, Italy, Kosovo, Lebanon, Turkey, Slovakia, Spain, UK, US [16,17,18,27,40,44,47,48,49,54,56,58,61]Biswas et al., Hajure et al., Li et al. [8,25,34]* Cross-sectional and cohort studies are categorized here according to the determinants that they found were associated with vaccine acceptance and the countries in which the study was conducted.


#### 3.2.5. Education

Higher education levels were associated with higher vaccine acceptance as highlighted in studies from the following countries: China (n = 2/5), Germany, Greece, Italy, the US, Turkey, and five reviews [8,20,23,25,27,29,34,43,46,53,55,60]. A Greek study found that 87.3% of HCWs with Master/Doctoral degrees and 88.4% with university-level degrees accepted the COVID-19 vaccine compared to 63.2% with high school level [20]. However, one Turkish study notes that vaccination rates were the highest not only in HCWs with Master/Doctoral/Specialization education levels (90.5%) but also among primary school graduates (87%) followed by bachelor (80.4%), high school (77.6%), and secondary school graduates (78.5%) [27]. College degree holders were 56% less likely to accept the vaccine compared to those with less than a college degree according to a US study [23].

#### 3.2.6. Chronic Diseases/Comorbidities

The presence of comorbidities or chronic diseases is a predictor of vaccine acceptance (n = 9/56) found in four reviews and five studies [8,17,18,23,25,29,34,36,57]. Interestingly, a study from the US finds that vaccine acceptors were more likely to have hypertension, and vaccine hesitant individuals were more likely to have asthma [23]. According to an Egyptian study, HCWs with chronic diseases were twice as likely to accept the COVID-19 vaccine compared to healthy HCWs [18]. Moreover, studies from Italy and the US each found that vaccine acceptance was associated with having diabetes, while one Saudi Arabian study reports that vaccine acceptance was lowest amongst those without chronic diseases [13,16,40].

### 3.3. Influenza Vaccination as a Factor for COVID-19 Vaccination

A substantial number of studies (n = 39/56) found that HCWs with a history of previous influenza vaccination are more likely to accept the COVID-19 vaccine [8,10,14,15,16,17,18,19,21,22,23,24,25,26,27,28,29,34,35,36,37,38,40,41,44,45,47,48,49,52,53,54,55,56,57,59,60,61,63]. Studies were predominantly from Europe (n = 15); most commonly from France (n = 4/5), Greece (n = 3/6), Italy (n = 3/4), Spain (n = 2/2), and Cyprus (n = 2/3) [14,15,19,21,22,36,38,40,45,47,48,49,54,56,63]. Nine studies were from the Middle East, such as Turkey (n = 4/4), Lebanon (n = 2/2), Azerbaijan, Palestine, and the UAE [17,24,26,27,37,41,44,52,61]. Other studies were from China (n = 3/5), the UK (n = 2/3), and the US (n = 2/2), in addition to six reviews [8,10,16,23,25,29,34,35,53,55,57,58,60]. Overall, Singapore had the highest influenza vaccination rate (92.2%) together with a high COVID-19 vaccine acceptance rate (94.9%) [30]. The lowest influenza vaccine uptake was reported in Turkey (17.0%) amongst pharmacists [44]. One study found that having received the influenza vaccine during the COVID-19 pandemic was linked to a 12% increase in the COVID-19 vaccine acceptance rate [41].

### 3.4. Factors and Barriers of Intention to Taking a COVID-19 Vaccine

Vaccine sentiments are determined partly by the value individuals put on the risks versus benefits of the COVID-19 disease and its vaccines. Positive and negative factors most commonly found to be associated with vaccine acceptance and the countries in which they were found in are highlighted in Table 3. Factors that positively influence uptake include psychological factors such as the fear of COVID-19 (n = 9/56), increased perception of risk (n = 11/56), and susceptibility (n = 7/56) [8,11,13,18,21,25,29,30,34,37,41,42,44,49,50,52,54,55,60,61,63]. The perception of having increased susceptibility is related to the knowledge of belonging to an at-risk group and having a fear of developing a severe COVID-19 infection [42,63]. The most common reasons that motivate HCWs to vaccinate were to protect others, such as family, patients, or close contacts (n = 15/56), and for self-protection (n = 11/56) [8,18,23,25,28,30,34,36,37,41,42,51,52,54,55,61,63]. HCWs have a sense of collective and social responsibility to reduce the number of COVID-19 cases, bring the pandemic under control, and obtain herd immunity (n = 11/56) [18,26,36,41,42,51,52,54,60,61,63]. Health literacy also seems to be an underlying factor in vaccination intent. For instance, being knowledgeable of the COVID-19 vaccines is linked to vaccine acceptance (n = 11/56), such as knowledge regarding acquired immunity, vaccine types, side effects, preventive measures, etc. [8,17,20,22,23,29,41,49,55,60,61]. In addition, trust was identified to be an important element in vaccine acceptance. Having trust in the recommendations provided by the government, health authorities, scientists, or public health experts (n = 10/56), and trust in the safety of the vaccine (n = 9/56), are associated with vaccine acceptance [25,28,34,45,49,53,57,60,61,64].

Several factors or barriers also negatively influence uptake, especially since the COVID-19 vaccines are new and their emerging technologies make them inherently a barrier. These include long- and short-term side effects of the vaccine (n = 32/56), its effectiveness (n = 18/56), novelty (n = 4/56), and quick development (n = 11/56) [8,10,12,14,16,19,20,22,25,26,27,28,29,32,34,36,37,39,41,42,45,49,50,51,52,54,55,56,57,58,61,63,64]. The lack of information (n = 10/56) that was clear, transparent, and informative of the safety and effectiveness of the vaccine was a barrier to vaccination [8,12,14,20,29,34,36,37,51,64]. Interestingly, previously confirmed COVID-19 infection is associated with negative attitudes towards vaccination (n = 16/56) [8,16,17,18,25,27,34,40,44,47,48,49,54,56,58,61]. This may contribute to a low self-perception of risk due to an inaccurate belief that acquired immunity against the virus will indefinitely protect from future COVID-19 infection [49]. A study that compared fear levels of COVID-19 found that vaccinated participants had significantly higher levels of fear than unvaccinated participants and were motivated by concerns about the disease itself compared to unvaccinated participants [54]. Furthermore, the low perceived risk of infection severity (n = 6/56) was itself identified as a barrier to COVID-19 vaccination [8,18,29,51,52,63]. Additionally, a lack of trust in the government and health authorities also seems to play a significant role in HCWs’ decision to vaccinate. Distrust in governments, health authorities, or pharmaceutical companies (n = 11/56) negatively affects vaccine confidence as vaccination campaigns are provided through these entities and may be further reinforced with misinformation and conspiracy theories [8,14,19,27,29,32,34,45,54,58,64].

### 3.5. Interventions

Providing a vaccine alone will not guarantee uptake of the COVID-19 vaccine and other evidence-based interventions must be implemented to ensure broader acceptance. Some studies found that HCWs believed that COVID-19 vaccination should be mandatory (n = 9), the greatest proportion (61.22%) being in a study from Italy and the lowest from Tunisia (5.4%) [8,13,20,23,24,29,35,36,46]. Mandatory vaccinations do not by themselves convince HCWs of the benefits of vaccines and diminish trust in healthcare organizations and vaccination campaigns [20,58]. Several studies identified a variety of interventions that should be implemented alongside vaccination campaigns in a multi-pronged approach. For instance, interventions that are focused on raising awareness regarding the safety, efficacy, and effectiveness of the COVID-19 vaccine can be implemented through educational campaigns, training sessions, or seminars (n = 18) to improve knowledge [8,14,18,19,20,23,27,29,33,34,35,36,43,49,52,55,57,64]. High-quality and transparent information can be disseminated through public health messaging and communication campaigns (n = 19) [14,16,17,20,22,23,24,27,28,33,36,40,48,49,54,57,62,63,64]. Social media marketing or mass media marketing (n = 7) can also be employed to improve uptake [11,16,19,28,37,54,64]. Directly addressing specific concerns of HCWs and following an individual-based approach during vaccination campaigns through targeted interventions (n = 11) towards sub-groups or non-vaccinated HCWs could lead to changes in sentiments about the COVID-19 vaccines [12,17,18,27,33,34,43,45,49,62,64]. Culturally sensitive interventions, improving access to vaccines, offering an incentive, building trust in vaccine manufacturers, and building resilience against misinformation were some of the other interventions recommended [8,11,12,16,23,27,28,29,33,37,43,54,57,59,64].

### 3.6. Vaccine Brands as a Factor for COVID-19 Vaccination

Vaccine acceptance may also be influenced by the type or brand of COVID-19 vaccines. Three studies investigated whether vaccine sentiments in HCWs were influenced by the different brands of COVID-19 vaccines which found that the most accepted vaccine was the mRNA vaccine from Pfizer/BioNTech [41,47,52]. For instance, one study found that 86% of HCWs that had received the vaccine had taken Pfizer/BioNTech, while only 8.8% and 5.3% accepted Sinopharm and Sputnik V vaccines, respectively [41]. When given multiple choices, HCWs that had yet not taken a COVID-19 vaccine, but were willing to accept one were most accepting towards Pfizer (71.9%), followed by Sputnik V (30.2%), Moderna (26.1%), AstraZeneca (21.4%), Janssen (12.3%), Sinopharm (12.1%), Sinovac (11.1%), and Novavax (2.1%) [41]. Similarly, a study from the UAE found that the most accepted vaccine was Pfizer/BioNTech (35%), followed by AstraZeneca (21%) and Sputnik V vaccines (4%) [52]. Alternatively, another study from France notes that 24% of HCWs were concerned about mRNA vaccines [42].

## 4. Discussion

This literature review was conducted to investigate and summarize HCWs’ sentiments towards COVID-19 vaccines, demographic determinants, their related factors, and barriers that they experience in their decision to vaccinate. Low vaccine acceptance impedes the control of infectious disease outbreaks and warrants investigation when introducing novel vaccines.

The proportion of HCWs accepting the COVID-19 vaccines varied significantly across regions and within regions across different countries. For instance, the overall highest and lowest vaccine acceptance was found within Europe in Italy (98.9%) and Germany (30%), respectively [15,19]. The German study was conducted among nurses and midwives, and our analysis of the literature revealed that vaccine acceptance was lowest amongst nursing professions [19]. This is concerning due to the prolonged contact between nurses and patients, contributing to a higher risk of contracting COVID-19, and a source of infection in nurses. A global scoping review among nurses reports that 23.4% and 18.3% of the nurses refused vaccines before the vaccine rollout (March 2020–December 2020) and after the COVID-19 vaccines were available (January 2021–May 2021), respectively [29]. The studies included in the aforementioned review were conducted during the emergency use authorization of COVID-19 vaccines, which may contribute to the lower refusal acceptance rate [29]. Additionally, the slight decrease in the refusal rate further supports SAGE’s conclusion that that vaccine sentiments are dynamic and can change as time progresses [5]. Increasing knowledge, popularity, the emergency use authorization of vaccines, and evidence of safety and effectiveness may have effected this change [29]. Nurses are frontline workers and advocates for public health, thus determining their sentiments is important to improve vaccine acceptance.

Several determinants were identified to be associated with vaccine acceptance and they may be interconnected. Perceptions of risk, vulnerability, and health-seeking behaviors may influence the decision to vaccinate [8,25,29,65]. For instance, older age was identified as a determinant of vaccine acceptance, which may be related to experience in a healthcare setting, level of education, and the higher chance of having comorbidities [8,29]. Age is a risk factor for COVID-19, putting older HCWs at a higher risk of hospitalization and death from COVID-19 [66]. Consequently, older HCWs may be motivated by their self-perceptions of vulnerability, and the knowledge of the benefits of vaccination [19,34]. Male HCWs are also more likely to accept the COVID-19 vaccines possibly due to a higher tendency to accept pharmaceutical measures [67]. In addition, research shows that males are more likely to become seriously ill due to COVID-19 compared to women [68]. Thus, being at a higher risk of adverse events could contribute to the higher uptake in males [34,65]. In addition, the type and level of education of HCWs may influence vaccine acceptance. One German study notes that higher education and medical training may allow individuals to discern between information and misinformation [43]. Education may be connected to health literacy, leading to a better understanding of vaccine efficacy and safety [43]. Thus, HCWs with lower education may have lower awareness and knowledge of vaccines and their benefits. Implementing interventions related to vaccine education to improve vaccine-related knowledge and health literacy will empower HCWs to make better informed vaccine-related decisions.

Ethnicity has not been widely studied as a determinant, possibly due to ethnically homogenous societies in several countries. However, ethnicity should be considered where possible, as comparatively distinct barriers may influence vaccine acceptance. They may include a history of discrimination, unethical research practices, institutional racism and discrimination, lack of representation in vaccine trials, and lack of prioritization within vaccine rollout [59]. A report by PAHO notes that the topic of trust must be addressed among Black ethnic groups to build confidence in vaccines [64]. This suggests that investigating vaccine acceptance among HCWs belonging to different ethnic groups may reveal different barriers, requiring interventions that are culturally sensitive and inclusive to improve vaccine uptake.

Above all, influenza vaccination was found to be the most significant factor in determining COVID-19 vaccine acceptance, as demonstrated by more than 65% of the included articles. Influenza vaccination in general may be an indicator of health-seeking behavior and acceptance of vaccines. Yet, there may be more concerns about the COVID-19 vaccine due to its novelty and perceived lack of safety. In addition, the influenza vaccine uptake rates may have increased during the pandemic. For instance, a Turkish study reports that the rate of pediatricians who received the seasonal influenza vaccine in 2020 was 39.1%, compared to 69.9% of those who wanted to get the vaccine in 2021 [24]. Another study from Italy reported there was a significant increase (∆% = 71.44) in influenza vaccination rate from the 2019–2020 campaign to the 2020–2021 campaign compared to campaigns before 2019 [46]. The risk of dual infections of influenza and COVID-19 during the influenza season may have positively influenced HCWs to receive the influenza vaccine [69].

While the determinants and factors of COVID-19 vaccine acceptance have been studied extensively in the literature, few studies assess whether different types or brands of COVID-19 vaccines influence HCWs decisions to vaccinate. Some studies have found that HCWs do put importance on the brand or type. A grey literature review notes that HCWs would be more willing to accept a vaccine if their preferred brand was available [64]. Another study from Lebanon notes that 77% of HCWs would be more positive about receiving the vaccines if they knew the type of vaccine [37]. In addition, COVID-19 vaccine acceptance may change with context and as evidence of the effectiveness of the different types of vaccines against variants emerges. For instance, a French study notes that the controversies of severe vaccine-related thrombosis from the AstraZenca COVID-19 vaccine impacted vaccine acceptance when the first dose of the vaccine was made available [48]. Interestingly, another French study notes that HCWs who had received at least one dose of the AstraZeneca COVID-19 vaccine were willing to receive a third dose as compared to those who had received at least one dose of the Moderna vaccine [47]. This is possible due to evidence that emerged at the time that the AstraZeneca vaccine is sub-optimal against the delta variant, which was the dominant variant at the time when the study was conducted [48]. This highlights the importance of conducting further studies to explore vaccine acceptance towards the different types or brands of COVID-19 vaccines and as new developments occur. One of the overarching findings was, to the best of our knowledge, no studies addressed public health professionals’ sentiments. Just as with HCWs, public health professionals have an important role in advising, promoting vaccinations, and addressing low vaccine acceptance [9]. This is uniquely so, as their primary role is to advocate for population health. Their knowledge and vaccination behavior is important to promote vaccinations among the general public [9]. This gap in current research warrants investigation so as not to miss the crucial opportunity to address concerns that may exist within this population.

A few limitations of this review should be considered when interpreting findings. First, only studies published in the English language were included, which may have introduced some bias. Second, we limited our search to the PubMed database; however, this database is well-known and comprehensive where many articles are indexed. Third, several studies were conducted before the rollout of vaccines and were assessed based on vaccination intent/willingness as opposed to vaccine uptake. Uptake rates may have changed as the pandemic continues to evolve and as confidence in the COVID-19 vaccines improves as time progresses. Finally, the studies included used a variety of questionnaires, methodology, outcomes, and analytical methods, which made the comparison of the results complex. For instance, several studies conducted assessments of the knowledge of HCWs of COVID-19, while some studies are based on self-assessment. A strength of our review was that a summative overview of the most common factors and barriers stratified was provided. Additionally, several key demographic determinants and sub-populations of HCWs likely to delay or refuse vaccines were identified, facilitating targeted interventions.

## 5. Conclusions

Determinants, such as male gender, older age, physician occupation, higher education levels, presence of comorbidities, and previous influenza vaccine history are personal determinants that are associated with high COVID-19 vaccination acceptance. Future research on the determinants of vaccine acceptance should investigate ethnicity and culturally sensitive interventions, as ethnicity is one of the more underappreciated determinants. In addition, further research is needed to explore why certain brands or types of vaccines are preferred which may reveal additional barriers. Educational interventions and transparent communications that address the concerns and worries held by HCWs can improve knowledge and adherence to additional doses and booster shots needed as the pandemic evolves and new variants emerge. Vaccine acceptance is multifaceted, and several barriers affect vaccination decision-making that can be addressed through multi-component interventions, tailored specifically for at-risk subgroups within HCWs.

## Figures and Tables

**Figure 1 vaccines-11-00311-f001:**
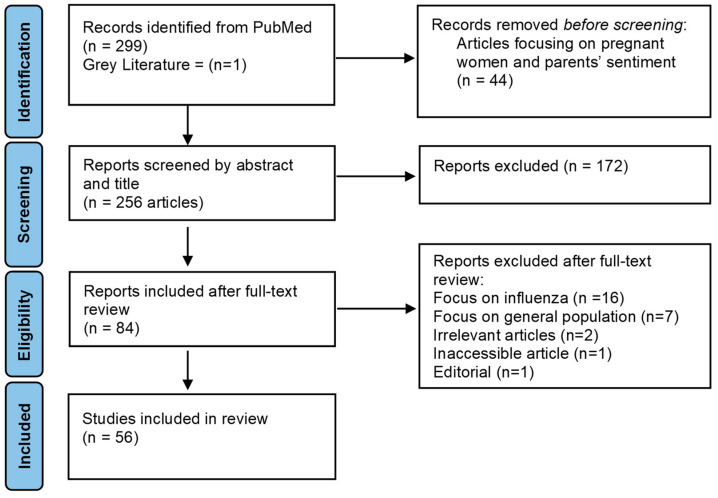
PRISMA flow chart depicting the screening and selection process.

**Table 1 vaccines-11-00311-t001:** Distribution of studies across countries and regions included in the review.

Regions	Country	Author	Number of Studies (n)
Europe	Albania	Patelarou et al. [49]	1
	Cyprus	Fakonti et al., Patelarou et al., Raftopolous et al. [19,49,51]	3
	Czech Republic	Štěpánek et al. [54]	1
	France	Gagneux-Brunon et al., Moirangthem et al., Navarre et al., Paris(a) et al., Paris(b) et al. [21,39,42,47,48]	5
	Germany	Nohl et al., Presotto et al. [43,50]	2
	Greece	Fotiadis et al., Galanis et al., Maltezou et al., Papagiannis et al., Patelarou et al., Raftopolous et al. [20,22,36,45,49,51]	6
	Italy	Di Gennaro et al., Di Valerio et al., Monami et al., Papini et al. [14,15,40,46]	4
	Kosovo	Patelarou et al. [49]	1
	Slovakia	Ulbrichtova et al. [56]	1
	Spain	Mena et al., Patelarou et al. [38,49]	2
	Switzerland	Zürcher et al. [63]	1
	UK	Abuown et al., Woolf (a) et al., Woolf (b) et al. [10,58,59]	3
Middle East	Azerbaijan	Doran et al. [17]	1
	Iraq	Al-Metwali et al. [11]	1
	Israel	Zaitoon et al. [62]	1
	Lebanon	Nasr et al., Youssef et al. [41,61]	2
	Palestine	Maraqa et al. [37]	1
	Saudi Arabia	Baghdadi et al. [13]	1
	Turkey	Gönüllü et al., Kara Esen et al., İkiışık et al., Okuyan et al. [24,26,27,44]	4
	UAE	Saddik et al. [52]	1
Africa	Egypt	Elkhayat et al. [18]	1
	Tunisia	Kefi et al. [28]	1
Asia	China	Leung et al., Kwok et al., Sun et al., Wang et al., Ye et al. [31,33,55,57,60]	5
	India	Ashok et al. [12]	1
	Singapore	Koh et al. [30]	1
North America	US	Do et al., Gatto et al. [16,23]	2
Reviews/Multi-Country Study/Report		Biswas et al., Hajure et al., Khubchandani et al., Leigh et al., Li et al., Luo et al., Shakeel et al., Report by PAHO [8,25,29,32,34,35,53,64]	8

**Table 3 vaccines-11-00311-t003:** Positive and negative factors commonly associated with vaccine acceptance and countries in which they were commonly found.

Positive Factors	Total Studies (n)	Country
Protection of Others	15	Cyprus, Czech Republic, France, Greece, US, Egypt, Tunisia, Singapore, Lebanon, Palestine, UAE
Control Pandemic/Disease	11	Cyprus, China, Czech Republic, Egypt, France, Greece, Lebanon, Switzerland, Turkey, UAE
Protect Themselves	11	Cyprus, Greece, Switzerland, Egypt, Tunisia, China, Singapore, Lebanon
Knowledge of COVID-19 and the Vaccine	11	Albania, Cyprus, Greece, Kosovo, Spain, US, China, Azerbaijan, Lebanon
Increased Perception of Risk	11	France, China, Singapore, Saudi Arabia, UAE
Trust in Government and Health Authorities	10	Albania, Cyprus, Greece, Kosovo, Spain, Tunisia, China, Lebanon
Increased Fear of COVID-19	9	Albania, Cyprus, Czech Republic, France, Greece, Kosovo, Spain, Lebanon, Palestine
Vaccine Safety	9	China, Greece, India, Azerbaijan, Saudi Arabia, Turkey
Increased Susceptibility to Disease	6	Switzerland, Egypt, China, Iraq, Lebanon, Palestine, Turkey
**Negative Factors**		
Vaccine Safety/Side Effects	32	Albania, Cyprus, Czech Republic, France, Germany, Greece, Italy, Kosovo, Slovakia, Spain, Switzerland, UK, US, Tunisia, Lebanon, China, Palestine, Turkey, UAE
Effectiveness and Efficacy	18	Albania, Cyprus, Czech Republic, Greece, Italy, Kosovo, Slovakia, Spain, UK, US, Tunisia, China, Lebanon, Palestine, Turkey
Previous COVID-19 Infection	16	Albania, Cyprus, Czech Republic, France, Greece, Italy, Kosovo, Slovakia, Spain, UK, US, Egypt, Azerbaijan, Lebanon, Turkey
Quick Vaccine Development	11	Cyprus, France, Greece, Italy, UK, India, Lebanon
Lack of Trust	11	Cyprus, Czech Republic, Greece, Italy, UK, Turkey
Lack of Information/Knowledge	10	Cyprus, Germany, Greece, India, Palestine
Misinformation	7	Switzerland, Egypt, Palestine
Infection not Severe/Harmful to Self	6	Cyprus, Switzerland, Egypt, UAE
Vaccine Content	4	France, Italy, Lebanon, Turkey
Vaccine Novelty	4	US
Wanting Others to take it First	4	US
Insufficient Time	3	Egypt, China
Vaccine Approval Process	2	Cyprus

## Data Availability

Data is available within the studies included for this review.

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
