# Peer review of "Determinants of Acceptance of COVID-19 Vaccination in Healthcare and Public Health Professionals: A Review"

_vaccines, 2023, doi:10.3390/vaccines11020311_

Round 1

Reviewer 1 Report

Although the paper refers to a subject which has been widely treated in the literature and does not bring any significant new contribution, it is quite well constructed and summarizes in a clear form the results of the studies included in the review. The paper may be accepted in the present form.

Author Response

Thank you very much for reviewing our manuscript. We appreciate the time taken to review our manuscript.

Reviewer 2 Report

Thank you for the opportunity to review this important review on COVID-19 vaccine acceptance among HCWs. This paper is a significant contribution, well organized, scientifically sound, and appropriately referenced. Minor revisions are suggested:

1. Abstract: Define the acronym “COVID-19”

 2. Introduction:

First sentence needs a citation.

Additionally, these sentences need to be cited. “Vaccines are the most effective intervention in controlling vaccine-preventable diseases. However, they have been increasingly scrutinized, resulting in low vaccine acceptance.” 

 3. Methods:

Define the timeframe of your Pubmed search. What was the beginning date of your search?

 4. Results

Was there any documented hesitancy related to different types or brands of COVID vaccine? (messenger RNA (mRNA) vaccines, viral vector vaccines, protein subunit vaccines, whole virus vaccines)

5. Conclusion: expand to include a statement about the need for further research regarding ethnicity and culturally sensitive interventions.

6. Figure 1: please verify screening and excluded #s.   299 + 1 – 44 – 99 – 16 – 9 – 1 does not equal 56.

My sincere appreciation to the authors for researching this important topic.

Author Response

Thank you for reviewing our manuscript. We really appreciate the valuable insights and suggestions provided and have made revisions accordingly.

Point 1: “Abstract: Define the acronym “COVID-19”

We have made the adjustments on Line 15 of the Abstract and Lines 35-36 in the Introduction as well. We have modified the abstract slightly to ensure we adhere to the requirements of the journal for a 200-word limit Abstract.

Point 2: “Additionally, these sentences need to be cited. “Vaccines are the most effective intervention in controlling vaccine preventable diseases. However, they have been increasingly scrutinized, resulting in low vaccine acceptance”

Thank you for this suggestion. We have provided citations accordingly. We decided to modify and expand this statement to “However, they have been increasingly scrutinized due to misinformation and social media platforms, resulting in low vaccine acceptance.” on Lines 43-44.

Point 3: “Define the timeframe of your Pubmed search. What was the beginning date of your search?”

We have added the beginning and end dates of our search on Line 78.

Point 4: “Was there any documented hesitancy related to different types or brands of COVID vaccine? (messenger RNA (mRNA) vaccines, viral vector vaccines, protein subunit vaccines, whole virus vaccines)”

Thank you for this important suggestion. We recognize how it is important to consider vaccine hesitancy in the context of the availability of multiple types/brands of COVID-19 vaccines and have revisited the literature. We incorporated your suggestion in the results section on Lines 300-313. As most of the studies included were conducted during the first few waves of the pandemic, there were very few articles that investigated the link between the type of COVID-19 vaccine and vaccine acceptance. As such, we have decided to also include a few lines in the discussion regarding this gap in research on Lines 381-399 and included a statement in the conclusion on Lines 429-430 for a need for further research on vaccine hesitancy related to the different types or brands.

Point 5: “Conclusion: expand to include a statement about the need for further research regarding ethnicity and culturally sensitive interventions.”

Thank you for this suggestion, we have incorporated your suggestions on Lines 426-429. In addition, we have added an additional point regarding ethnicity so that the discussion ties in with the conclusion on Lines 365-368.

Point 6: “Figure 1: please verify screening and excluded #s. 299 + 1 – 44 – 99 – 16 – 9 – 1 does not equal 56.”

Thank you for bringing this to our attention. We apologize for the mistakes in our numbers and have revisited our screening process and the excluded number of articles. We previously categorized excluded articles together in the flow chart but have further expanded in the revised version of the flow chart providing specific numbers for each exclusion e.g. “Editorial (n=1), Irrelevant articles (n=2)” to avoid confusion.

Reviewer 3 Report

This is a timely and important study in the field of vaccine behavior conducted by Ghare et al to assess determinants including male gender, older age, physician occupation, and other comorbidities tied with high COVID-19 vaccination acceptance.

there are a few comments that can be used to enhance the submitted work:

This work is highly timely and fits well with the newly arising crisis of COVID in China, and the way this work is performed looks like a systematic review which is excellent:

I advise the authors to register this work with PROSPERO and resubmit it as a Systematic review in the title

The work would benefit to discuss the vaccination pattern in China and how the sentiment can be affected by the country’s health policy, it would be important to add references to the pattern in China and how the health workers would react to vaccination due to the country policy and how they perceive it.

the work should have a future perspective of the implication of the outcome of these studies and can public health intervention be applied to affect this vaccination approval

Overall, I enjoyed reading this article.

Author Response

Thank you for your valuable recommendations, we have provided below our response to each suggestion.

Point 1: “I advise the authors to register this work with PROSPERO and resubmit it as a Systematic review in the title”

Thank you very much for your advice. We have tried to conduct this study as systematically as possible while also ensuring that results are described as comprehensively and concisely as possible. While planning this study, we realized that due to time constraints and capacity, we would not be able to do a very comprehensive study that assesses the heterogeneity of included articles, an assessment of the risk of bias, and a quality assessment of the included articles. As such, we prefer to call this a literature review that provides an overview of the current research and the gaps that currently exist.

Point 2: “The work would benefit to discuss the vaccination pattern in China and how the sentiment can be affected by the country’s health policy, it would be important to add references to the pattern in China and how the health workers would react to vaccination due to the country policy and how they perceive it.”

Thank you for your suggestions. The authors had a prior discussion when planning and designing the review and decided to make this a global study rather than focus on any specific country. As such, we did not have any specific inclusion/exclusion for any country and tried to provide a broad overview of the studies conducted globally. We are thankful for your recommendation, and we will consider focusing on COVID-19 vaccine sentiments specifically in China in a future study.

Point 3: “The work should have a future perspective of the implication of the outcome of these studies and can public health intervention be applied to affect this vaccination approval”

Thank you for this suggestion, we appreciate this insightful suggestion. We have made modifications on Lines 388-392 that state the need for future research on public health interventions to improve vaccine uptake.